# Non-Visible Light Data Synthesis: A Case Study for Synthetic Aperture Radar Imagery

## Abstract

Large-scale pre-trained image generation models such as Stable Diffusion and Imagen have achieved a nearly perfect synthesis of regular images. We explore their "hidden" application in non-visible light domains, taking Synthetic Aperture Radar (SAR) data for a case study. For SAR, due to the inherent challenges in capturing satellite data, acquiring ample training samples is problematic. For instance, for a particular category of ship in the open sea, we can collect only a few dozen SAR images which are too limited to derive effective ship recognition models. If pre-trained regular image models can be adapted to generate diverse SAR data, the problem is solved. In preliminary experiments, we found that directly fine-tuning these models with existing SAR data cannot generate meaningful or novel SAR data. The main challenge is the difficulty in capturing the two primary differences between SAR and regular images: structure and modality. To address this, we propose a 2-stage low-rank adaptation method, and we call it 2LoRA. In the first stage, the model is adapted using aerial-view regular image data (whose structure matches SAR), followed by the second stage where the base model from the first stage is further adapted using SAR modality data. Particularly in the second stage, we introduce a novel prototype LoRA (pLoRA), as an improved version of 2LoRA, to resolve the class imbalance problem in the original SAR dataset. For evaluation, we employ the resulting generation model (e.g., ControlNet+pLoRA) to synthesize additional SAR data. This augmentation, when integrated into the training process of SAR recognition models, yields notably improved performance for minor classes. *Codes are in the Appendix.*

## 1 Introduction

Large-scale pre-trained generative models, such as Stable Diffusion (SD) (Rombach et al., 2021), Imagen (Saharia et al., 2022), GLIDE (Nichol et al., 2021), and ControlNet Zhang & Agrawala (2023) (based on SD) can generate realistic and diverse regular images given textual or structural prompts. Their success comes from the capacity of learning a robust and generalizable representation space from billions of web images (Schuhmann et al., 2022; Kwon et al., 2022). In this paper, we focus on synthesizing data for an uncommon data modality Synthetic Aperture Radar (SAR) whose spectrum is outside human's visible light range (400-700 nm) but has wide applications in marine safety, environmental protection, and climate studies. In our preliminary study with the popular open-sourced SD model, we found that fine-tuning it on SAR (whose data is often insufficient) tends to overfit that modality and "forget" the general features learned from its original pre-training. Our other observation is that straightforwardly employing the popular "anti-forgetting" domain adaptation method LoRA (low-rank decomposition) (Hu et al., 2021) does not solve the problem. The main challenge is the difficulty in capturing the two primary differences: structure and modality, between regular images and SAR images. Fine-tuning or low-rank decomposition on SD fails to capture the necessary feature transformations arising from the two differences. In Figure 1 (a), columns 4 and 5 display the synthesis results from the fine-tuned SD and LoRA-based SD, respectively, both of which are suboptimal. The SAR recognition result (F1 score), using corresponding synthesized SAR for data augmentation, is given below each column as a quantitative reference.

So, the question arises: why, given such a large domain gap, do we still consider using the pre-trained knowledge from regular images? Our motivation comes from an initial empirical observation below. When we don't perform any fine-tuning or LoRA adaptation on SD and directly use the SD

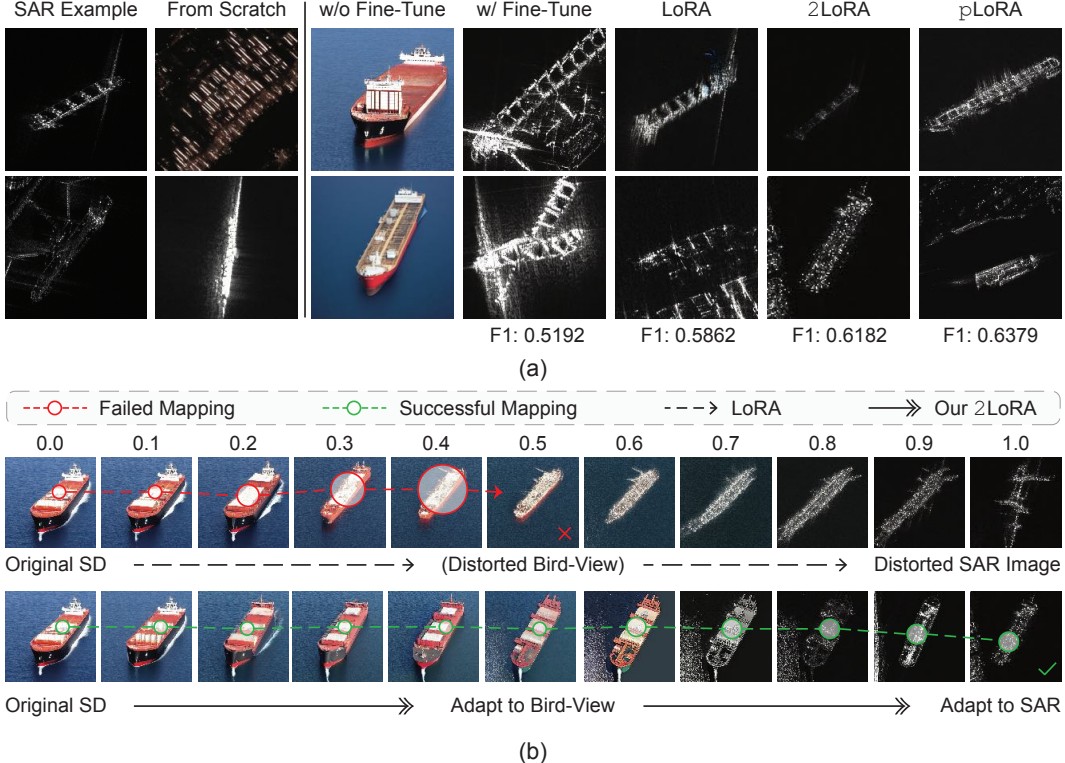

Figure 1: SAR imagery synthesis for ship class "tanker", by different methods in (a), and detailed comparison between LoRA and our $2$LoRA in (b). Methods in (a) include: training an LDM model with SAR data from scratch; inference with a frozen SD; fine-tuning SD; learning a LoRA with SAR data; our $2$LoRA; our $p$LoRA. The F1 score of the "tanker" ship is given under each column, as a quantitative reference for the utility of synthesized images. In (b), the scale of applying LoRA (or $2$LoRA) on SD is given on top of each column. Please note that all images are generated with the same text prompt "A SAR image of a tanker ship". Please zoom in for a better view.

features (extracted for SAR data) to train a SAR recognition model, we found that this model can easily outperform existing ones which are usually small-scale, trained from scratch, or with a simple pre-training on ImageNet (Ridnik et al., 2021). The possible reasons are three-fold. 1) SAR data contains noise (due to atmospheric conditions, sensor noise, etc.), and the SD model inherently has denoising properties. 2) SAR data often exhibit high variability (due to factors such as different sensors, acquisition times, and weather conditions), and the SD model based on large-scale pre-training is robust to such variations. 3) SAR data have spatially coherent structures (e.g., urban areas, water bodies, and forests), and the diffusion process in SD respects spatial coherence, making SD adept at preserving these structures while extracting meaningful features. Therefore, we believe if the SD model with its general vision knowledge can be further adapted to the SAR domain, e.g., it can synthesize diverse and high-quality SAR samples, it will address the data scarcity issue of SAR.

As mentioned, directly adapting SD with SAR images is not working due to the large domain gap. We dive into the detailed visualization of failure cases of LoRA in the first row of Figure 1 (b), taking "tanker ship" as an example. A LoRA scale of 0 implies the absence of LoRA (on top of SD), producing a conventional RGB image by prompting "tanker ship" for SD. As the scale adjusts to 0.5, noticeable distortions emerge, rendering a blurry deck. With a further increase to a scale of 1.0, the distortion becomes egregious, resulting in the very wrong structure of "tanker ship". So, the finding is that the transition from a regular view (scale = 0) to an aerial view (approximately at scale = 0.5) fails in SD, and this failure propagates to result in a totally flawed SAR image (when the scale reaches 1.0). An intuitive solution is that enhancing the SD's synthesis ability on aerial-view images may prevent the distortion at the first stage (i.e., from a regular view to an aerial view). It is also a soundable solution since one of the key differences between regular images and SAR images is the structure which is fortunately not significantly different between aerial-view images and SAR

images. More fortunately, we can leverage optical remote-sensing (ORS) images whose large-scale datasets (Zhang et al., 2021; Ding et al., 2021) are available, to achieve it.

We thus propose a 2-stage LoRA[1] approach to adapt SD from regular imagery to SAR imagery indirectly, and we call it 2LoRA. In the first stage, the model is learned to adapt from its regular view to the aerial view without changing its data modality, by training an ORS LoRA module on ORS datasets. In the second stage, the model is further adapted from RGB modality to SAR modality, by training a SAR LoRA module on SAR datasets. Particularly, in the second stage, we introduce a novel prototype LoRA, dubbed pLoRA, as an improved version of 2LoRA, to resolve the class imbalance problem in SAR datasets. In other words, 2LoRA struggles with minor classes such as "dredger ship" due to data limits, while pLoRA does not. For pLoRA, we first cluster all SAR training samples on the feature space regardless of their classes, assuming each cluster captures the primary attributes of a specific SAR imagery prototype, e.g., "long and slender hull", "wide deck" or "angular bow". Then, we use the samples in each cluster to train an individual pLoRA, and weighted sum all pLoRAs to substitute the vanilla SAR LoRA (in 2LoRA), improving the synthesis of minor classes. Using pLoRA, we see a more coherent transition from regular-view RGB images to aerial-view SAR images, in the second row of Figure 1(b).

Our technical contributions in this paper are two-fold: 1) a pioneer work of leveraging large-scale pre-trained generation models for synthesizing non-visible light images, i.e., transferring the semantic knowledge pre-learned in regular images to diversity the training data which has a significant domain gap; and 2) a novel 2LoRA approach that addresses the domain adaptation challenges from regular images to SAR images, and its improved version pLoRA to solve the class imbalance problem in the original SAR dataset. For evaluating 2LoRA and pLoRA, we employ the resulting models to synthesize additional SAR data for minor classes. This data augmentation, when integrated into the training process of SAR recognition models, yields notably improved performance.

## 2 RELATED WORKS

**Data Augmentation with Synthetic Images.** Natural images synthesized by pre-trained diffusion models have been validated to be effective in augmenting regular image datasets (Zhang et al., 2022; He et al., 2022; Azizi et al., 2023). One work by He et al. (2022) combines large models like GLIDE (Nichol et al., 2021), T5 (Raffel et al., 2019) and CLIP Radford et al. (2021) to generate high-quality natural images. Their methods effectively tackled few-shot, zero-shot, and data imbalance problems in original datasets. Another work by Rombach et al. (2021) shows that images generated by Denoising Diffusion Probabilistic Models (DDPMs) can enhance ImageNet pre-trained networks. To increase semantic variance in generated images, the GIF (Zhang et al., 2022) is designed by adding a marginal perturbation in the latent space of Latent Diffusion Models (LDMs). However, all of the existing works generate natural images that share similar visual representations with the pre-training images of the large models. The domain gap between the application and pre-training images is small. Besides, most of the target datasets (to be augmented) don't really have long-lasting data shortage problems as their images are mainly from daily life or can be collected from ordinary cameras, compared to the non-visible light data on specific objects (*e.g.*, "ships" in maritime monitoring) which are hard to collect. Our focus in this work is a non-visible light data synthesis, taking SAR imagery as a study case. SAR imagery has an inherent large domain gap with natural images regarding both structure and modality. The SAR ship classification task faces long-lasting data scarcity issues due to the military implications of maritime monitoring.

**Conventional Data Augmentation.** Data augmentation methods are widely used to improve the diversity of visual datasets, thereby boosting the generalization ability of trained models (Shorten & Khoshgoftaar, 2019). Popular strategies include erasing (Zhong et al., 2017), image manipulation (Wang et al., 2017), cutmix (Yun et al., 2019), and search-based methods (Cubuk et al., 2019). These strategies, as validated by Zhang et al. (2022), typically operating on existing images with manually specified rules, are limited to local pixel-wise modifications without introducing novel content or unseen visual concepts. In comparison, we perform data augmentation from a data synthesis perspective. We thus introduce a new way of leveraging the pre-trained knowledge (*e.g.*, that in SD) to generate various training samples in new data domains (*e.g.*, SAR in our case).

---

[1]We don't consider fine-tuning SD to avoid catastrophic forgetting of SD's pre-trained knowledge.

**Generative Model Adapation.** Adapting pre-trained generative models, such as GAN (Karras et al., 2020; Brock et al., 2018) and Stable Diffusion Rombach et al. (2021), aims to tranfer the knowledge of these models to synthesize new concepts or out-of-distribution data. Existing methods can be roughly classified into two categories: *concept-level* and *domain-level*. *Concept-level* adaptation modifies the model knowledge to moderate new visual concepts (unseen during pre-training), such as for generating new objects, styles, or certain spatial structures. Within the GAN paradigm, some works (Ojha et al., 2021; Gal et al., 2022b) fine-tune the entire generator using regularization techniques. In contrast, some other works try to optimize the crucial part of generator (Kim et al., 2022; Alanov et al., 2022), or introduce a lightweight attribute adaptor before the frozen generator and a classifier after the frozen discriminator (Yang et al., 2021). On top of the SD model, Dream-Booth (Ruiz et al., 2022) and Textual Inversion (Gal et al., 2022a) try to generate new objects in existing scenes. ControlNet (Zhang & Agrawala, 2023) and DragDiffusion Shi et al. (2023) enable the image generation to be conditioned on a certain spatial structure. HyperDreamBooth (Ruiz et al., 2023) customizes the generated image to show a certain human face. All these adaptation methods are based on the assumption that the "SD backbone" could readily generate any unseen concept, i.e., the concept has been seen by SD during its pre-training with regular images. *Domain-level* adaptation aims to adapt a pre-trained generative model to a new image domain absent (or very rare) from its pre-training data, such as medical images. Chambon et al. (2022) validate that fine-tuning SD with carefully selected hyperparameters could lead to realistic lung X-ray images. Khader et al. (2022) find that, compared to GAN, the diffusion model is more capable of encompassing the diversity of medical images. Unlike these works, we focus on the SAR domain, and find that fine-tuning or employing a direct low-rank decomposition on SD fails to capture the needed feature transformation, due to the significant domain gap between regular and SAR data as well as the data insufficiency of SAR.

## 3 PRELIMINARIES

**Stable Diffusion.** We implement our method on top of a large-scale pre-trained diffusion model: Stable Diffusion (SD) (Rombach et al., 2021). We chose it because of three aspects. 1) SAR data contains noise, and the SD model inherently has denoising properties. 2) SAR data often exhibit high variability, and the SD model based on large-scale pre-training is robust to such variations. 3) SAR data have spatial coherent structures, and the diffusion process in SD respects spatial coherence. Specifically, SD is a text-to-image model that incorporates a diffusion process in the latent space of a pre-trained autoencoder (Esser et al., 2020). In SD, a denoising U-Net is trained to fit the distribution of latent codes, and it is conditioned on the textual embeddings extracted through a text encoder CLIP (Radford et al., 2021) via cross-attention. During inference, SD performs iterative reverse diffusion on a randomly sampled noise to generate an image that faithfully adheres to the input text. Given a data pair $(x, \tau)^2$, where $x$ is an image and $\tau$ is text prompt, the learning objective for SD is to minimize a denoising objective as:

$$\mathcal{L}(x, \tau; \theta) = \mathbb{E}_{\mathcal{E}(x), \tau, \epsilon \sim \mathcal{N}(0,1), t}[\|\epsilon - \epsilon_\theta(z_t, t, \psi(\tau))\|_2^2], \tag{1}$$

where $z_t$ is the latent feature at timestep $t$, $\psi$ is pre-trained CLIP text encoder, $\mathcal{E}$ is pre-trained VQGAN encoder, and $\epsilon_\theta$ is the denoising U-Net with learnable parameter $\theta$.

**Low-Rank Adaptation.** Our implementation of domain adaptation from natural images to SAR is based on low-rank adaptation (LoRA) Hu et al. (2021). LoRA was initially proposed to adapt large-language models to downstream tasks. It operates under the assumption that during model updating, parameter updating is usually sparse. It thus introduces a low-rank factorization of the parameter changes, i.e., $\Delta\theta := B \cdot A$. Here, $\theta \in \mathbb{R}^{d \times k}$ represents the parameters of pre-trained model (*e.g.*, SD in our case), and $B \in \mathbb{R}^{d \times r}$ and $A \in \mathbb{R}^{r \times k}$ denote low-rank factors, with $r \ll \min(d, k)$. The updated parameters $\theta'$ are thus given by $\theta' = \theta + \Delta\theta = \theta + B \cdot A$. Injecting multiple concepts can be realized by training multiple LoRA modules (each for a single concept) and combining them with a weighted sum as $\theta' = \theta + \sum_i w_i \Delta\theta_i$, where $w_i$ denotes combination weights.

---

[2]To avoid confusion with notations in the downstream classification task, we denote text prompts as $\tau$, classfication labels as $y$.

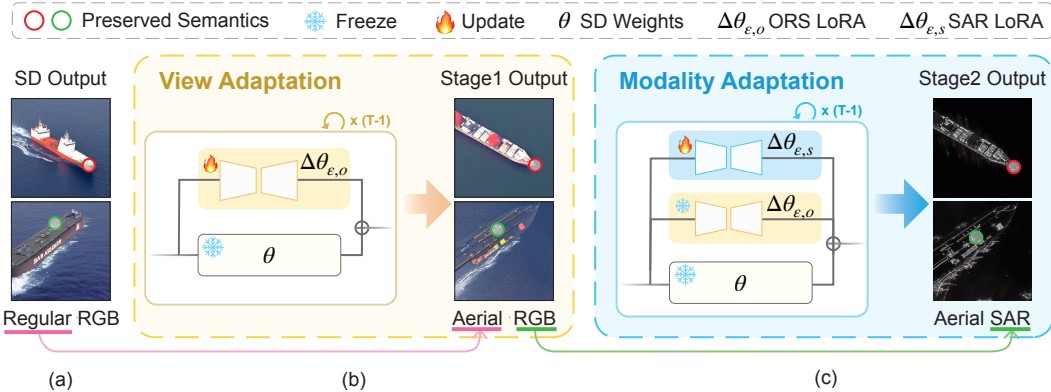

Figure 2: **The training pipeline of 2LoRA.** We train LoRA modules by 2 stages. In the first stage, *i.e.*, view adaptation, we train an ORS LoRA $\Delta W_o$ on top of SD's cross-attention layers. In the second stage, *i.e.*, modality adaptation, we train a SAR LoRA $\Delta W_s$ to further adapt to SAR modality (from RGB modality). The learning is on top of a frozen SD-v1.5. The "Regular View" images are generated by SD without any LoRA. The "Aerial RGB" and "Aerial SAR" images are the outputs of the two stages of LoRAs, respectively. Particularly, we highlight the preserved semantics, *e.g.*, angle of the bow (in red circles) and yellow oil pipe (in green circles) on the generated images.

## 4 SD ADAPTATION AND SAR DATA AUGMENTATION

We synthesize high-utility SAR data by adapting SD with the proposed 2LoRA (or the improved version pLoRA), to augment minor classes in SAR datasets. Without loss of generality, we use a SAR ship classification task (that has open-sourced datasets) as an eventual evaluation of the proposed methods. More specifically, 2LoRA has two stages, including view adaptation and modality adaptation (respectively in Sections 4.1 and 4.2). pLoRA particularly solves the class imbalance problem in original SAR datasets in an unsupervised manner. For data augmentation (Section 4.3), we propose to adapt a structure-conditional SD, i.e., ControlNet (Zhao et al., 2023), using pLoRA, and feed the adapted model with diverse edge conditions "borrowed" from large-scale ORS datasets.

### 4.1 VIEW ADAPTATION

Figure 2 (a) shows that SD without adaptation generates only regular-view RGB images. To make it capture the view of SAR, in the first stage, we adapt it from regular views to aerial views. We learn an ORS LoRA on top of a frozen SD, as in Figure 2 (b). To this end, we use open-sourced aerial-view ORS datasets Zhang et al. (2021); Ding et al. (2021). SD requires text prompts during training. So, we first elaborate on how we generate prompts on these datasets.

**Prompt Construction.** During training, SD associates the visual knowledge in images with the semantics in text prompts, through a cross-attention mechanism. Low-quality text prompts introduce ambiguity to this association. Given the fact that large-scale ORS datasets such as ShipRSImageNet (Zhang et al., 2021) and DOTAv2 (Ding et al., 2021) provide only categorical annotations, the question is how to generate high-quality prompts efficiently from such labels. Previously, this was manually solved by human prompt engineers. In this work, we propose to leverage Large Language Models (LLMs) such as GPT-4 and LLAMA to automate this engineering process.

Our pipeline is shown in Figure 3. Given a dataset with annotations of bounding boxes, ship category, data source, and camera settings, we employ GPT-4 to extract visual components (*e.g.*, deck, hull, and cabin) and factors (*e.g.*, ship correlations, weather conditions, ship orientations) that exhibit distinct visual representations in aerial views. Then, we let GPT-4 generate detailed visual descriptions, which encompass four aspects: 1) spatial correlation between ship instances; 2) image visibility (*e.g.*, foggy/clear, and dark/bright); 3) sailing direction of the ship; 4) visual attributes of ship components (*e.g.*, texture, shape, size, and positions). Finally, we request GPT-4 to extract keyword lists from these descriptions (*i.e.*, by removing uninformative natural language words, such as "is", "and"). We use these keywords as the final text prompts. By using them, we build the data

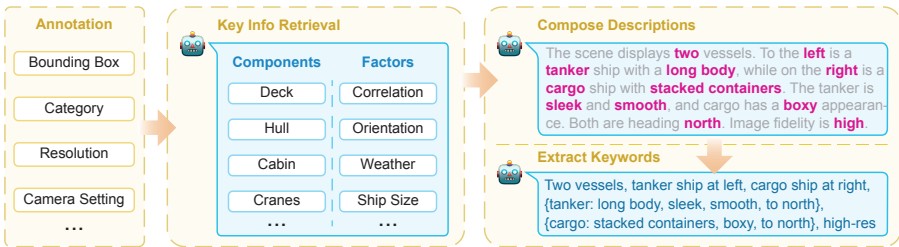

Figure 3: **Prompt construction.** We use GPT-4 (OpenAI, 2023) to construct high-quality prompts for ORS data. The robot emoji indicates the GPT-4-0314 LLM engine.

triplets $\{(\text{ORS image, category, prompt})\}$ and denote them as $\{(x_o, y_o, \tau_o)\}$, with subscript "o" for ORS. *Please kindly note that we put our communication history with GPT-4 in the Appendix.*

**ORS LoRA.** We learn an ORS LoRA on the training data $\{(x_o, \tau_o)\}$, to let the model "understand" aerial views. The direct fine-tuning of SD can be formulated with loss $\mathcal{L}(x, \tau; \theta)$ (Eq. 1) as:

$$\arg\min_{\Delta\theta} \mathcal{L}(x_o, \tau_o; \theta + \Delta\theta), \tag{2}$$

where $\theta$ denotes learnable parameters in the denoising U-Net (of SD), $\theta + \Delta\theta$ denotes updated parameters, $+$ means directly modifying $\theta$ with gradient descent, and $\Delta\theta$ indicates the difference. We assume this $\Delta\theta$ is low-rank decomposable (Hu et al., 2021). That means, it can be parameterized by a smaller set of parameters $\varepsilon$, *i.e.*, $\Delta\theta := \varepsilon$. Therefore, we can search for an optimal low-rank $\varepsilon$ with a parameter-efficient learning objective as: $\arg\min_{\varepsilon} \mathcal{L}(x_o, \tau_o; \theta + \varepsilon)$, where $+$ means skip-connecting a layer parameterized by $\varepsilon$ to $\theta$, following the original architecture design of LoRA.

SD contains a sequence of attention layers, and $\varepsilon$ in LoRA is designed as a sequence of light networks. Each network consists of two linear layers with a hidden dimension $r$, corresponding to the "rank" in low-rank decomposition. As shown in Figure 2 (b), each LoRA network is attached to one attention layer of SD. For real implementation, we apply LoRA networks on the cross-attention layers (*i.e.*, on Q, K, V linear projection layers), because the research by Hu et al. (2021) has shown that tuning the attention layers in transformers is sufficient and efficient.

We call the learned LoRA networks an "ORS LoRA" module in the following, and represent it by $\varepsilon_o$ where "o" is for ORS. During *inference*, this module outputs parameter updates $\varepsilon_o$ for each cross-attention layer. The updated network $\theta'$ is thus as $\theta' = \theta + w_1 \varepsilon_o$, where $w_1$ is the strength of the module and $+$ means skip-connection.

## 4.2 MODALITY ADAPTATION

**Prompt Construction.** Unlike ORS datasets, SAR ship datasets (Zhang et al., 2021; Ding et al., 2021) don't contain extensive information on camera setup, weather, or ship orientation, but only categories. Therefore, we use a simple template-based prompt "a SAR image of {category} ship" as the text input to SD during SAR LoRA training. We denote the data triples {SAR image, category, prompt} as $\{(x_s, y_s, \tau_s)\}$, where "s" stands for SAR.

**2LoRA.** In the second stage, we adapt SD *w/* ORS LoRA from RGB modality to SAR modality. We achieve this by learning a SAR LoRA on top of them, as shown in Figure 2 (c). Specifically, we freeze the model of SD *w/* ORS LoRA and train a SAR LoRA $\varepsilon_s$ on the training data $\{(x_s, \tau_s)\}$.

During the inference of 2LoRA, we combine the learned LoRA modules as:

$$\theta_{2\text{LoRA}} = \theta + w_1 \varepsilon_o + w_2 \varepsilon_s \tag{3}$$

where $w_1$ and $w_2$ are hyperparameters denoting the strengths of LoRA modules. Please note that the sum of $w_1$ and $w_2$ is not necessarily to be 1. As pointed by Hu et al. (2021), a higher LoRA strength approximates a higher learning rate. In our experiments, we empirically allow $0.5 < w_1 + w_2 < 2$, because we observed clear image distortion beyond this range.

**pLoRA.** 2LoRA can successfully adapt SD from the regular image domain to the SAR domain, while we observe bias problems from the results of 2LoRA due to the data imbalance in SAR training data (even with resampling). As shown in the first row of Table 2, the generation quality

of $2$LoRA significantly biases to the major class "cargo" with the lowest FID score ($0.8479$). The minor classes with fewer training samples, *i.e.*, "tanker" and "dredger", suffer from poorer synthesis (*i.e.*, higher FID values as $1.1024$ and $0.9670$, respectively).

We tackle this by clustering the SAR dataset into several groups each representing a visual prototype/attribute of SAR ships, e.g., "wide deck" or "angular bow". Each prototype includes samples from various SAR ship classes, reducing bias due to class imbalance. To this end, we first extract the feature from SAR images $\{x_s\}$ with a pre-trained SAR classifier $f$ (*e.g.*, ResNet50): $\mathbf{F} = f(x)$, where $\mathbf{F}$ is the extracted features. We apply $K$-Means clustering on $\mathbf{F}$ to get $K$ groups. Each group captures an identical visual attribute such as "small ships" or "foggy weather" (see Appendix Figure A1). Using the feature group indices, we divide the original SAR dataset into $K$ groups. Each group is then used to train an individual LoRA, following the same method as aforementioned (for ORS LoRA). We get $K$ prototype LoRAs (dubbed pLoRA). We denote their parameters as $\varepsilon_p, p \in [1, K]$.

Using pLoRA for *inference* means we combine ORS LoRA $\varepsilon_o$ and pLoRA $\varepsilon_p$ with a weighted sum:

$$\theta_{\mathrm{pLoRA}} = \theta + w_o \Delta\varepsilon_o + \sum_{p=1}^{K} w_p \varepsilon_p. \tag{4}$$

Here, the weights $\mathbf{w} = [w_1, w_2, ...w_K]$ are critical to avoid categories too dependent on biased knowledge. When generating images for a specific category, we calculate the sample ratio of this category relative to the total training samples of the cluster. This produces a "bias score" per cluster. A cluster with a higher bias score suggests less bias towards other classes, leading us to assign a greater weight to this cluster. Given cluster distribution $N(C, P)$ where $C = \{c_i\}$ stand for categories, and $P = \{p_j\}$ for clusters. When generating image for $i$-th category, the bias score of $i$-th category in $j$-th cluster is calculated as $b_{i,j} = N(c_i, p_j) / \sum_i N(c_i, p_j)$. We thus can get a bias vector for $i$-th category as $\mathbf{b_i} = [b_{i,1}, b_{i,2}, ...b_{i,K}]$. We then use the L1 normalization of $\mathbf{b}$ as our pLoRA weights, *i.e.*, $\mathbf{w} = \frac{\mathbf{b}}{||\mathbf{b}||_1}$. *Please kindly refer to the Appendix for more details of computing weights $w_p$ and the advantage justification of pLoRA over 2LoRA.*

## 4.3 DATASET AUGMENTATION

Without loss of generality, we use a SAR ship classification task, which has open-sourced datasets for benchmarking, as an eventual evaluation for our synthesis-based data augmentation approach. We first assume a small-scale real SAR dataset of fine-grained ships $\mathcal{D} = \{(x, y)\}$ is given. Here, a sample pair contains a SAR image $x$ with its category label $y$, *e.g.*, "tanker". Our goal is to synthesize a new dataset $\mathcal{D}' = \{(x', y')\}$, such that the classifier trained with $\mathcal{D} \cup \mathcal{D}'$ has improved performance, especially for minor classes.

The key points are 1) the data distribution in synthetic dataset $\mathcal{D}'$ should align with the original dataset $\mathcal{D}$ in the semantic feature space, *e.g.*, the ship structure keeps the same between real and synthetic "tanker" images, and 2) $\mathcal{D}'$ should contain higher semantic diversity than $\mathcal{D}$, *e.g.*, new contexts or "tanker"-related attributes can be generated on an existing "tanker" sample.

Specifically, for *point 1*, we ensure the structure coherence between real and synthesized images by using a conditional SD called ControlNet (Zhang & Agrawala, 2023). ControlNet constrains the generated images to faithfully follow a fixed spatial structure, such as contour, depth, or pose. The contour version is used for our implementation. It uses a shadow copy of the denoising U-Net to learn contour-to-image mapping and adds this mapping as a plug-in to SD. During inference, given a reference ship image $x$, we feed its contour $c$ (Canny Edge (Canny, 1986)) into ControlNet and text prompt $\tau$ into SD. The generated image $x'$ will have the same contour $c$ (as the reference image $x$). In our implementation of 2LoRA (or pLoRA), we also fine-tune the contour-based ControlNet on ORS (stage 1) or SAR (stage 2) datasets. This fine-tuning is optional and does not significantly affect the final results. For *point 2*, we introduce more diversity by taking ORS instances as reference images. For example, ShipRSImageNet (Zhang et al., 2021) initially has 161 "tanker" instances. After eliminating crowded or incomplete instances, 47 edge maps remain, which we use as reference when synthesizing SAR "tanker" images. Consequently, we possess 362 contours for "tanker". For each category during inference, 3,000 images are produced. We then randomly choose some for augmentation. It's important to note two things: 1) for FID score evaluation, only the SAR ship contours are used to ensure a consistent real-to-synthetic image pairing, and 2) when comparing image generation techniques for recognition tasks, images are created using the same contour set.

## 5 EXPERIMENTS

**SAR ship dataset.** We introduce a novel dataset designed for the fine-grained classification of SAR ships. Existing datasets often suffer from issues like low resolution and limited test samples. Upon reviewing various fine-grained ship classification datasets (as detailed in Appendix Table 3), we found that only FUSAR-Ship (Hou et al., 2020) and SRSDD (Lei et al., 2021) meet the resolution requirements ($\leq$ 10m) suitable for training a

Table 1: **Statistics of the FU-SRS dataset.**

| Index | Name | Train | Deci | Val | Test | Total |
|---|---|---|---|---|---|---|
| 0 | Cargo | 3,890 | 389 | 484 | 772 | 5,146 |
| 1 | Other | 1,710 | 171 | 213 | 216 | 2,139 |
| 2 | Fishing | 814 | 81 | 102 | 161 | 1,077 |
| 3 | Tanker | 315 | 32 | 40 | 59 | 414 |
| 4 | Dredger | 242 | 24 | 30 | 52 | 324 |
| | Total | 6,971 | 696 | 869 | 1,260 | 9,100 |

generative model. We wanted to confidently evaluate models using these datasets. To ensure enough test samples for every ship class, we combined the above two datasets. Categories with fewer than 10 test samples were labeled as "others". The resulting combined dataset is named FU-SRS. Furthermore, to define a stronger data-shortage problem, we derive a "Deci" split from the training set of FU-SRS where "Deci-" means one-tenth, by randomly sampling $\frac{1}{10}$ samples. The statistics of these two dataset settings are shown in Table 1. Following the Pareto principle (also known as the 20–80 principle), we define categories with $\leq 20\%$ training samples as "minor class", *i.e.*, the "fishing", "tanker", and "dredger". *Please note that this dataset is open-sourced anonymously for paper review. More details are in the Appendix.*

**ORS ship dataset.** As mentioned in Section 4.3, we constructed ORS image-prompt pairs to train our ORS LoRA, based on two ORS datasets: DOTAv2 (Ding et al., 2021) and ShipRSImageNet Zhang et al. (2021). For DOTAv2, we use its ship or harbor instances, and build prompts with the simple template "ORS, optical remote sensing, <ship/harbor>". For ShipRSImageNet, we use the method in Section 4.1 to build informative prompts. *More details are in the Appendix.*

**Evaluation metrics.** We use Frechlet Inception Distance (FID) for evaluating image synthesis quality and F1-Score for data augmentation performance. The FID score is calculated between synthetic and real SAR images, and we use ResNet50 pre-trained on our FUSRS dataset as the feature extractor. We use ControlNet (*w/* or *w/o* our methods) to synthesize images given the real images as reference (for extracting contours), to ensure that the synthetic images are comparable to the reference. Please note that this applies to all comparing FID scores. The F1 scores evaluate the ship classifiers after using our data augmentation. We double the training samples of minor ship categories, *i.e.*, "fishing", "tanker", and "dredger". For example, for "tanker" ship with 315 real samples, we synthesize additional 315 samples. To evaluate the classifiers more robustly, we average the top 5 models' F1 scores to derive a final score.

Table 2: **Comapre `2LoRA`, `cLoRA`, and `pLoRA`, regarding their synthetic image quality (FID) and data augmentation performance (F1).** They have the same ORS LoRA but different SAR LoRAs: `2LoRA` has a single 16-rank SAR LoRA; `cLoRA` has four 4-rank LoRAs respectively trained with the data of "fishing", "tanker", "dredger" and the others; and `pLoRA` is our final model, i.e., four 4-rank prototype LoRA. "†" denotes using the same $w_p = 0.25$ for all prototype LoRAs.

| Name | Cargo | Fishing | | Tanker | | Dredger | | Average | |
|---|---|---|---|---|---|---|---|---|---|
| | FID↓ | FID↓ | F1↑ | FID↓ | F1↑ | FID↓ | F1↑ | FID↓ | F1↑ |
| 2LoRA | 0.8479 | 0.9301 | 0.7063 | 1.1024 | 0.6182 | 0.9670 | 0.8214 | 0.9619 | 0.7153 |
| cLoRA | **0.8402** | 0.9272 | **0.7197** | 1.0971 | 0.6018 | 0.9615 | 0.8182 | 0.9565 | 0.7132 |
| pLoRA† | 0.8591 | **0.9107** | 0.7148 | 0.9346 | 0.6195 | **0.9118** | 0.8246 | **0.9041** | 0.7196 |
| pLoRA | 0.8474 | 0.9344 | 0.7153 | **0.9262** | **0.6379** | 0.9390 | **0.8448** | 0.9118 | **0.7327** |

**Ablation study.** We compare three modality adaption strategies: `2LoRA` has a single 16-rank SAR LoRA; `cLoRA` has four 4-rank LoRAs respectively trained with the data of "fishing", "tanker", "dredger" and the others; and `pLoRA` is our final model, i.e., four 4-rank prototype LoRA. We have some interesting observations from Table 2. 1) `cLoRA` shows better FID than `2LoRA` but is not better for F1-Score, suggesting that though `cLoRA` might produce more similar images to real ones (indicated by lower FID), its synthesized images are not useful for data augmentation which requires data diversity. 2) Compared to `2LoRA` and `cLoRA`, `pLoRA` exhibits better F1-Score values in most

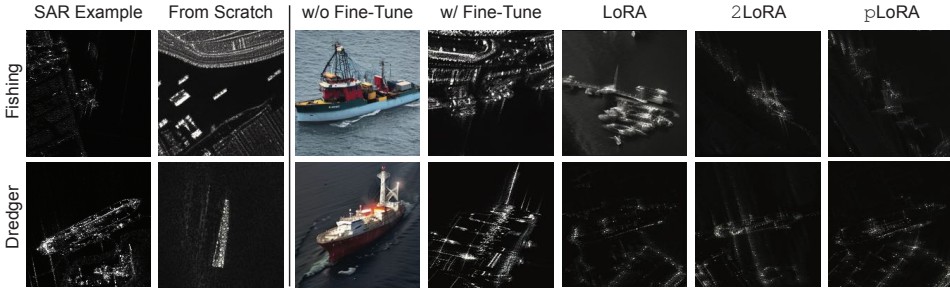

Figure 4: **Qualitative comparison.** SAR image synthesis for minor classes "fishing" and "dredger", with prompt "A SAR image of {category} ship". The backbone is SD, and the first column of SAR samples is for reference but not the ground truth (as there is no ground truth).

cases, especially when compared to $2$LoRA, suggesting its capability to alleviate the class imbalance problem and generate useful data for augmentation. 3) Comparing pLoRA$^{\dagger}$ with pLoRA, we see that although the FID of pLoRA is consistently higher than pLoRA$^{\dagger}$, it demonstrates a noticeable improvement in F1-Score.

Table 3: **Compare different generation methods.** We address the data imbalance issue by doubling the training samples for minor classes, and we compare different strategies of generation. The "Resample" means additional samples are resampled from original training data. We use ship classification precision, recall, and F1-Score (abbreviated as P, R, and F) as measures.

| Name | Split | Cargo | | | Other | | | Fishing | | | Tanker | | | Dredger | | |
|---|---|---|---|---|---|---|---|---|---|---|---|---|---|---|---|---|
| | | P | R | F | P | R | F | P | R | F | P | R | F | P | R | F |
| Resample | Deci | 0.8936 | 0.9249 | 0.9090 | 0.7056 | 0.7546 | 0.7293 | 0.6547 | 0.5652 | 0.6067 | 0.4082 | 0.3390 | 0.3704 | 0.6429 | 0.5192 | 0.5745 |
| Fine-Tune | Deci | 0.8938 | 0.9262 | 0.9097 | 0.7156 | 0.7454 | 0.7302 | 0.6454 | 0.5652 | 0.6026 | 0.3600 | 0.3051 | 0.3303 | 0.5909 | 0.5000 | 0.5417 |
| 2LoRA | Deci | 0.9058 | **0.9339** | **0.9196** | **0.7253** | **0.7824** | **0.7528** | **0.6812** | **0.5839** | **0.6288** | **0.4400** | 0.3729 | 0.4037 | **0.6744** | 0.5577 | 0.6105 |
| pLoRA | Deci | **0.9111** | 0.9028 | 0.9070 | 0.6721 | 0.7685 | 0.7171 | 0.6357 | 0.5093 | 0.5655 | 0.4237 | **0.4237** | **0.4237** | 0.6000 | **0.6923** | **0.6429** |
| Resample | Train | 0.8996 | **0.9637** | 0.9306 | **0.8199** | 0.8009 | **0.8103** | 0.7273 | 0.5963 | 0.6553 | 0.6957 | 0.5424 | 0.6095 | **0.8864** | 0.7500 | 0.8125 |
| Fine-Tune | Train | 0.8696 | 0.9585 | 0.9119 | 0.8021 | 0.7130 | 0.7549 | 0.7266 | 0.5776 | 0.6436 | 0.6000 | 0.4576 | 0.5192 | **0.8864** | 0.7500 | 0.8125 |
| LoRA | Train | 0.8796 | 0.9560 | 0.9162 | 0.8022 | 0.6759 | 0.7337 | 0.7266 | 0.5776 | 0.6436 | 0.5965 | 0.5763 | 0.5862 | 0.7778 | 0.8077 | 0.7925 |
| 2LoRA | Train | **0.9185** | 0.9495 | **0.9338** | 0.7788 | **0.8148** | 0.7964 | 0.8080 | 0.6273 | 0.7063 | **0.6667** | 0.5763 | 0.6182 | 0.7667 | 0.8846 | 0.8214 |
| pLoRA | Train | 0.9172 | 0.9184 | 0.9178 | 0.7113 | 0.7870 | 0.7473 | **0.8110** | 0.6398 | **0.7153** | 0.6491 | **0.6271** | **0.6379** | 0.7656 | **0.9423** | **0.8448** |

**Quantitative comparisons.** In Table 3, we compare the image generation methods for tackling the class imbalance problem. We adopt resampling as our baseline and evaluated four generative methods: 1) vanilla fine-tuning SD on SAR, 2) training a single SAR LoRA, 3) our proposed $2$LoRA, and 4) our proposed pLoRA. To evaluate each method's performance on a smaller data scale, we also conducted experiments on the "Deci" split of FU-SRS dataset (which is 1/10 of the training set, see Table 1). On FU-SRS, pLoRA and 2LoRA outperform the baseline methods on the minor classes, especially the recall value (pLoRA outperforms Resample by $0.192$ for dredger, $0.085$ for tanker, and $0.044$ for fishing). This suggests the model tends to classify more samples as minor classes. Though the precision may decrease, the F1 score increases stably. On "Deci" FU-SRS, both pLoRA and 2LoRA consistently surpass the baseline methods. Notably, pLoRA appears predominantly focused on enhancing the performance of the minor classes, whereas 2LoRA exhibits improvement for both minor and major classes.

**Qualitative results.** Similar to Figure 1 (a), we show synthetic examples of minor classes in Figure 4. Fine-tuning SD or learning a single LoRA faces view distortion problems (*i.e.*, the images are not in aerial views). In contrast, our $2$LoRA and pLoRA generates realistic SAR images.

## 6 CONCLUSIONS

We explored the hidden potentials of large-scale pre-trained image generation models in non-visible light domains. We revealed some intriguing findings that led us to the 2-stage low-rank adaptation method $2$LoRA and its improved version pLoAR. Future research could include exploring the scalability of our approach to handle even larger datasets of SAR or more diverse modalities related to SAR, and optimizing the adaptation process for other non-visible light domains such as magnetic resonance imaging and infrared imaging.

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
