## A  APPENDIX

Anonymous code and FU-SAR dataset for review is available at `https://anonymous.4open.science/r/ICLR24-4617`.

Table A1: **SAR ship datasets since 2017.** We use our method to tackle small-class problems in SAR ship classification, and only FUSAR-Ship and SRSDD-v1.0 meet our criteria. Datasets marked by "†" means it is deprecated due to low resolution, and "∗" means insufficient categories.

| Dataset | Year | Category | Instances | Width (px) | Resolution (/px) |
|---|---|---|---|---|---|
| OpenSARShip2 (Li et al., 2017)† | 2017 | 16 | 19,360 | 30–120 | 22m |
| SAR-Ship-Dataset (Wang et al., 2019)∗ | 2019 | 1 | 59,535 | 256 | 3m–25m |
| AIR-SARShip-2.0 (Wang et al., 2023)∗ | 2020 | 1 | 461 | 1000 | 1m, 3m |
| FUSAR-Ship (Hou et al., 2020) | 2020 | 15 | 6,358 | 512 | ≥0.5m |
| HRSID (Wei et al., 2020)∗ | 2020 | 1 | 16,951 | 800 | 0.5m, 1m, 3m |
| LS-SSDD-v1.0 (Zhang et al., 2020)∗ | 2020 | 1 | 6,015 | 16,000 | 20m |
| Official SSDD (Zhang et al., 2021a)∗ | 2021 | 1 | 2,456 | 190–160 | 1m–15m |
| SRSDD-v1.0 (Lei et al., 2021) | 2021 | 6 | 2,884 | 512 | 1m |
| RSDD-SAR (Congan et al., 2022)∗ | 2022 | 1 | 10,263 | 512 | 2m–20m |
| xView3-SAR (Paolo et al., 2022)† | 2023 | 2 | 243,018 | 512 | 10m |

Table A2: **Sample numbers of FUSAR dataset.** The FUSAR dataset faces problems of insufficient test samples and vaguely defined classes. Categories marked by "†" means it is deprecated due to insufficient test samples, and "∗" means being vaguely defined. Results are shown in classification F1-Score, and "IM21K" indicates pre-trained ResNet50 on the IM21K dataset.

| Category | Cargo | DiveVessel† | Dredger | Fishing | HighSpeedCraft† | LawEnforce† | Other∗ | Passenger† |
|---|---|---|---|---|---|---|---|---|
| Test Sample | 325 | 1 | 14 | 120 | 3 | 3 | 412 | 6 |

| Category | PortTender†∗ | Reserved†∗ | SAR† | Tanker | Tug† | Unspecified∗ | WingInGrnd†∗ | |
|---|---|---|---|---|---|---|---|---|
| Test Sample | 1 | 9 | 2 | 34 | 9 | 14 | 2 | |

### A.1  DATASETS

**FUSRS dataset.** As listed in Table A1, most of the SAR ship datasets either only contain a broad "ship" category without subcategories, or are of low resolution (≥ 10m). Only FUSAR-Ship and SRSDD v1.0 datasets met our resolution and category criteria. However, as shown in Table A2, FUSAR-Ship had issues with insufficient test samples and vaguely defined categories. To address this, we merged the ship categories from SRSDD v1.0 and FUSAR-Ship, removed categories with fewer than 10 test samples (to the "others" category), and named this new dataset as FUSRS (Table 1). Our experiments showed that with vote-ensembling, the test results' standard deviation on FUSRS significantly converged to ±3 in F1-Score, indicating that FUSRS can serve as a robust benchmark. We will open-source this FUSRS dataset and benchmarking codes.

**ORS dataset.** To train the ORS LoRA, we need an ORS image-caption dataset. Due to the lack of such datasets in existing ORS research, we collected one from the DOTAv2 and the ShipRSImageNet datasets. We introduce the preprocessing of DOTAv2 and ShipRSImageNet respectively here: The DOTAv2 dataset is a large-scale aerial detection dataset, which contains 50,356 ship/harbor instances. We crop out the ship/harbor instances as image patches, and label these patches with simple captions like "ORS, optical remote sensing, <ship/harbor>". The ShipRSImageNet dataset is an ORS ship dataset for detection, and its annotation files contain extensive information (*i.e.*, the coordinates of ships, the correlation between ships, the weather conditions, etc.). We extract key information from the detection annotation files and use GPT-4 to organize that information into image descriptions. This ORS image-caption dataset will be open-sourced.

## A.2 IMPLEMENTATIONS

**Detailed implementations.** LoRA training is trained with batch size 32. ORS LoRA was trained for 200 epochs and SAR LoRA for 100, with a cosine annealing scheduler starting at a learning rate of $1 \times 10^{-3}$. We fine-tune ControlNet for 30 epochs with batch size 4, using a learning rate of $4 \times 10^{-5}$. We keep other training settings the same as ControlNet. For the recognition model, we fine-tune SAR data based on an ImageNet-21K pre-trained ResNet50. We use 4 NVIDIA RTX 3090 or Tesla V100 GPUs with a batch size 128. We use SGD with a learning rate of 0.1, momentum of 0.9, and weight decay of $1 \times 10^{-4}$. The scheduler was cosine annealing with a warm-up of 500 iterations from $1 \times 10^{-4}$, and the model was trained for 100 epochs.

**Generation configs** During inference, we empirically set hyper-parameters for LoRA generation as: CFG=7.5, sampling steps 30, ControlNet combination weight $0.4$ when training epochs $\geq 10$, and weight $0.8$ when training epochs $\leq 10$. We adopt ControlNet "balanced mode" as the original implementation in GitHub.

Table A3: **Rank of 2LoRA.** We compare how different LoRA ranks affect the generated image quality in the second stage (adaptation stage).

| Rank | Params (M) | Cargo | | Fishing | | Tanker | | Dredger | |
|---|---|---|---|---|---|---|---|---|---|
| | | FID↓ | F1↑ | FID↓ | F1↑ | FID↓ | F1↑ | FID↓ | F1↑ |
| 4 | 0.0797 | 0.9042 | **0.9351** | 0.9290 | 0.7088 | 1.0714 | **0.6296** | 1.1296 | 0.8214 |
| 8 | 0.1594 | 0.8504 | 0.9338 | **0.8970** | 0.7092 | 1.1492 | 0.5981 | 1.0079 | 0.8142 |
| 16 | 0.3188 | **0.8479** | 0.9338 | 0.9301 | 0.7063 | **1.0227** | 0.6182 | **0.9670** | **0.8214** |
| 32 | 0.6377 | 0.8519 | 0.9344 | 0.9094 | **0.7138** | 1.0907 | 0.6038 | 1.0714 | 0.8000 |

## A.3 THE VALUE OF RANK.

The rank of the LoRA component controls the number of learnable parameters. As shown in Table A3, we compare the FID and F1-Score of 2LoRA generated images with different LoRA ranks. Considering the overall performance of three minor classes, we set the rank as 16 in our experiment.

## A.4 JUSTIFICATION

We formulate $w_p$ as follows: given cluster distribution $N(C, P)$ where $C = \{c_i\}$ is the categories, and $P = \{p_j\}$ is clusters. When generating image for $i$-th category, the bias score of $i$-th category in $j$-th cluster is calculated as $b_{i,j} = N(c_i, p_j)/\sum_i N(c_i, p_j)$. For example, in Table A4, bias score for "tanker" ship in cluster $p_1$ is $239/3153$, and in cluster $p_2$ is $4/436$. We thus could obtain a bias score vector $\mathbf{b} = [b_j]$. For example, in Table A4, the bias vector for the "tanker" ship is $\mathbf{b} = [239/3153, 4/436, 23/569, 89/2590]$. We then use the L1 normalization of $\mathbf{b}$ as our LoRA combination weights.

Table A4: **Clusters with K=4.**

| Category | $p_1$ | $p_2$ | $p_3$ | $p_4$ |
|---|---|---|---|---|
| Cargo | 1,513 | 153 | 417 | 1,299 |
| Other | 876 | 128 | 76 | 843 |
| Fishing | 354 | 149 | 45 | 368 |
| Tanker | 239 | 4 | 23 | 89 |
| Dredger | 171 | 2 | 8 | 91 |
| All | 3,153 | 436 | 569 | 2,690 |

We justify that pLoRA is less prone to bias problems than 2LoRA. Take the "tanker" ship as an example. In the 2LoRA, the training samples of the "tanker" ship only occupy $4.5\%$ of the dataset. If learning a shared SAR LoRA (as in 2LoRA), the knowledge of "tanker" ship will be severely overwritten by major classes such as "cargo": as shown in Table 2, the "tanker" ship got a high FID of $1.1024$ as compared with "cargo" whose FID is $0.8479$.

Suppose we set $K$=4 in pLoRA. In that case, we decompose the training data into 4 prototypes, respectively containing $a=[239, 4, 23, 89]$ "tanker" instances. We first calculate the bias score of "tanker" ship in each cluster by sample number/cluster sample number: for cluster $p_1$, the bias score for "tanker" is $239/3153=0.076$; for cluster $p_2$, the bias score is $4/436=0.009$. Similarly, we can get $23/569=0.040$ and $89/2690=0.033$ for $p_3$ and $p_4$ respectively. The resulting bias score vector is $b=[0.076, 0.009, 0.040, 0.033]$. Compare the 3-rd and the 4-th clusters: though "tanker" ship got

| Cluster 1 Small Ships | Cluster 2 Ships in Harbor | Cluster 3 Foggy Weather | Cluster 4 Moving Ships |
|---|---|---|---|

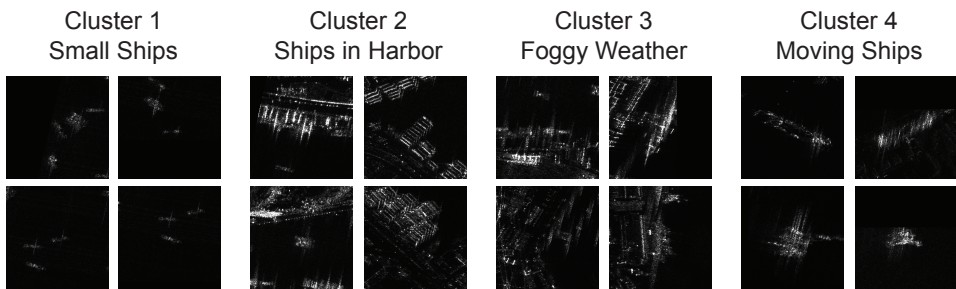

Figure A1: **Illustration of Clusters.** The four clusters captures distinct visual representations, such as "fast-moving", "small" ships or "foggy" weather.

more training samples in $4$-th prototype training ($89>23$), the $4$-th cluster suffers more from data imbalance problem for "tanker" ($0.033<0.040$). Thus, we should rely more on the $3$-rd prototype, and reduce dependence on the $4$-th prototype. We apply L1 normalization on $b$ to achieve this. The resulting normalized bias-score $\hat{b}=[0.47, 0.05, 0.26, 0.20]$, where the $3$-rd prototype is put a clearly higher score. We take the $\hat{b}$ as our prototype weights $w_p$.

## A.5 PROMPT CONSTRUCTION

We show GPT-4 chat history at this anonymous link (`https://anonymous.4open.science/r/ICLR24-4617`).

## A.6 ILLUSTRATION OF CLUSTERS

We show clustered samples in Figure A1.