# OpenReview forum: "Non-Visible Light Data Synthesis: A Case Study for Synthetic Aperture Radar Imagery"
_ICLR.cc/2024/Conference — ICLR 2024 Conference Withdrawn Submission_

### Official Review · Reviewer_8JRb · 2023-10-19

**Soundness:** 2 fair
**Presentation:** 2 fair
**Contribution:** 1 poor
**Rating:** 3
**Confidence:** 4

**Summary:**

The paper presents a method to generate synthetic SAR images via finetuning diffusion models to overcome the limited availability of annotated SAR data for classification problems. The domain gap between RGB and SAR images includes both a viewpoint variation, being SAR aerial and a modality variation related to the statistics of SAR data. The method uses two-stages of LoRA finetuning to bridge this gap, separately addressing the two issues.

**Strengths:**

The paper addresses an important problem in the context of using deep learning in remote sensing, which is the scarcity of annotated data, especially for certain acquisition modalities like SAR. Using synthetically generated images can be a sensible way of performing data augmentation, that has already been shown effective in other domains.

**Weaknesses:**

The overall novelty of work is limited. The methodology is based on well-known concepts such as LoRA and ControlNet. The two-stage approach to adaptation, while important for problem, is minor and of interest only for a niche audience. Overall, the clarity of the paper can be improved. For example, the presentation of the method is often intertwined with experimental details.

Experimental evaluation is also problematic. The FID metric is not suitable for SAR images, since it is a distance in the latent space of an ImageNet-trained neural network, thus having a large domain gap (some works point out that the ImageNet prior affects evaluation even of RGB images, nevermind SAR, see Kynkäänniemi et al. "The Role of ImageNet Classes in Fréchet Inception Distance"). The F1 score of the downstream classification task is a relevant metric but the results are difficult to understand. For instance, Table 2 does not report the F1 performance of the baseline without augmentations.

**Questions:**

What data have been used to train the modality adaptation part? Is it the train split of FUSAR?

---

### Official Review · Reviewer_68N7 · 2023-10-31

**Soundness:** 2 fair
**Presentation:** 2 fair
**Contribution:** 2 fair
**Rating:** 5
**Confidence:** 4

**Summary:**

In this paper, the authors introduced a 2-stage LoRA (Label-agnostic OoD Representations Adaptation) approach, termed 2LoRA, which aimed to adapt semantic knowledge from regular imagery to synthetic aperture radar (SAR) imagery indirectly. The first stage involved training an ORS LoRA module on ORS datasets to adapt from a regular view to an aerial view without changing data modality. In the second stage, a SAR LoRA module was trained on SAR datasets to further adapt from RGB modality to SAR modality. Notably, the second stage introduced a prototype LoRA (pLoRA) to address class imbalance issues in SAR datasets. The pLoRA clustered SAR training samples based on their features, with each cluster representing a specific SAR imagery prototype, and individual pLoRAs were trained for each cluster. These pLoRAs were weighted and combined to enhance the synthesis of minor classes, resulting in a more coherent transition from regular-view RGB images to aerial-view SAR images.

**Strengths:**

The strengths of this research lie in its innovative approach to addressing the challenges of adapting semantic knowledge from regular imagery to synthetic aperture radar (SAR) imagery. The 2LoRA method and its enhanced version, pLoRA, offer a novel solution to overcome class imbalance issues in SAR datasets, particularly for minor classes. The paper's pioneering use of large-scale pre-trained generation models for synthesizing non-visible light images is a notable technical contribution, allowing the transfer of pre-learned semantic knowledge from regular images to SAR data despite significant domain gaps.

**Weaknesses:**

A comprehensive comparison is essential to substantiate the efficacy of the proposed approach. Additionally, the authors' reliance on a custom dataset limits the ability to assess the method's effectiveness without validation on publicly available datasets. Furthermore, there are concerns regarding stability, including potential dependencies in prompt construction and the generation of detailed visual descriptions by GPT-4.

**Questions:**

The paper lacks a comprehensive comparison with existing methods. Can the authors provide a more detailed discussion or experimentation comparing their proposed 2LoRA and pLoRA methods with other state-of-the-art domain adaptation techniques for SAR imagery? It would strengthen the paper's contribution and help assess the novelty and effectiveness of their approach.

The authors designed their own dataset, which may have inherent biases. Have the authors considered testing their methods on publicly available SAR datasets to provide a more generalized evaluation? This would enhance the external validity of their findings.

One key aspect that the authors could explore is the practical implementation of the generated data from their proposed methods in real-world applications. This would not only provide a tangible demonstration of the utility of their techniques but also make the research more relevant and applicable to various domains, such as remote sensing, defense, or environmental monitoring.

---

### Official Review · Reviewer_zzFN · 2023-10-31

**Soundness:** 3 good
**Presentation:** 3 good
**Contribution:** 3 good
**Rating:** 5
**Confidence:** 4

**Summary:**

This paper proposes a 2-stage low-rank adaptation (2LoRA) method for non-visible light data synthesis. The first stage adapts the model using aerial-view regular image data, and the second stage further adapts the model using SAR modality data. Further, improving the 2LoRA  to resolve the class imbalance problem in the original SAR dataset, named prototype LoRA (pLoRA). The experimental results show that the proposed method can improve the performance for minor classes.

**Strengths:**

Data synthesis is important and valuable for non-visible light data, which is hard to collect large-scale dataset.

**Weaknesses:**

1. The proposed methods do not show significant performance improvement when using the full traning dataset. In practical, we always will employ all data to train a model rather than only employ 10% data. However, when employing the full training data, the resample strategy is simple and comparable with the proposed method.
2. The pLoRA is not effective. According to the experiments, pLoRA performs worse than 2LoRA in most cases.

**Questions:**

1. Why do not provide the F1 for Cargo in Table 2?
2. Why the performance of the proposed method is different in Tables 2 and 3?

---

### Official Review · Reviewer_xb7k · 2023-10-31

**Soundness:** 3 good
**Presentation:** 4 excellent
**Contribution:** 4 excellent
**Rating:** 8
**Confidence:** 4

**Summary:**

The authors present a method fro SAR data - with augmentation of data and an improved training process.
The paper is well-written without errors.

**Strengths:**

The paper is well written and the background sound. A large amount of content has been presented in a short number of pages very well.
I find no plagiarism either.

**Weaknesses:**

The paper may not be easily accessible to non-experts as the paper is written very high-level. Due to space limitations, however, this is ok.

**Questions:**

In the abstract, code is referred to - this need not be in the abstract, and should rather be in the application section.